# Identification of Critical Immune Regulators and Potential Interactions of IL-26 in *Riemerella anatipestifer*-Infected Ducks by Transcriptome Analysis and Profiling

**DOI:** 10.3390/microorganisms12050973

**Published:** 2024-05-12

**Authors:** Paula Leona T. Cammayo-Fletcher, Rochelle A. Flores, Binh T. Nguyen, Bujinlkham Altanzul, Cherry P. Fernandez-Colorado, Woo H. Kim, Rajkumari Mandakini Devi, Suk Kim, Wongi Min

**Affiliations:** 1College of Veterinary Medicine & Institute of Animal Medicine, Gyeongsang National University, Jinju 52828, Republic of Korea; cammayopaula@gmail.com (P.L.T.C.-F.); floresrochellea@gmail.com (R.A.F.); thanhbinhcnty@gmail.com (B.T.N.); bvjbvjka@gmail.com (B.A.); woohyun.kim@gnu.ac.kr (W.H.K.); kimsuk@gnu.ac.kr (S.K.); 2Department of Veterinary Paraclinical Sciences, College of Veterinary Medicine, University of the Philippines Los Baños, Los Baños 4031, Philippines; cherryfernandez0423@gmail.com; 3Department of Veterinary Microbiology, College of Veterinary Sciences & Animal Husbandry, Central Agricultural University (1), Jalukie 797110, India; mandaraaj@gmail.com

**Keywords:** ducks, *Riemerella anatipestifer* infection, spleen, RNA-seq, IL-17A, IL-26

## Abstract

*Riemerella anatipestifer* (RA) is an economically important pathogen in the duck industry worldwide that causes high mortality and morbidity in infected birds. We previously found that upregulated IL-17A expression in ducks infected with RA participates in the pathogenesis of the disease, but this mechanism is not linked to IL-23, which primarily promotes Th17 cell differentiation and proliferation. RNA sequencing analysis was used in this study to investigate other mechanisms of IL-17A upregulation in RA infection. A possible interaction of IL-26 and IL-17 was discovered, highlighting the potential of IL-26 as a novel upstream cytokine that can regulate IL-17A during RA infection. Additionally, this process identified several important pathways and genes related to the complex networks and potential regulation of the host immune response in RA-infected ducks. Collectively, these findings not only serve as a roadmap for our understanding of RA infection and the development of new immunotherapeutic approaches for this disease, but they also provide an opportunity to understand the immune system of ducks.

## 1. Introduction

A *transcriptome* is defined broadly as the whole set of transcripts from the genome of a cell and their quantities, encompassing messenger RNA (mRNA), non-coding RNA, and small RNA for a defined developmental stage and physiological or pathological condition [1,2]. Conventionally, the quantification of transcriptomes is achieved through approaches using hybridization- and sequence-based technologies such as microarrays or Sanger sequencing of cDNAs or expressed sequence tag libraries. Transcriptomic analysis has been employed to evaluate the functional elements of a genome and the molecular subtyping of cells and tissues. Technological advancements with the use of high-throughput sequencing technologies for transcriptomics, such as RNA sequencing (RNA-seq), have revolutionized our understanding of transcriptomes [1]. Currently, RNA-seq is the gold standard of transcriptome analysis and has been used to characterize biological markers and quantitate gene expression differences between tissues [2,3]. It has also been extensively utilized to monitor and accurately track gene expression and gene expression changes, characterize novel molecular subgroups, and discover modifications (i.e., fusion genes) to understand disease development and progression or treatment and drug responses [1,2,3].

*Riemerella anatipestifer* (RA), formerly known as *Pasteurella anatipestifer,* is a Gram-negative bacterium from the *Flavobacteriaceae* family that is considered one of the most economically impactful infectious diseases affecting domestic fowl species worldwide. It mainly infects ducklings and contributes to strain-dependent mortality and morbidity rates of up to 90% and 100%, respectively, in infected ducks or stunted growth in convalescent birds [4,5,6]. RA infection can be acute or chronic, with acute infection being more common in ducks younger than 8 weeks of age [7]. Infection is often referred to as new duck disease, riemerellosis, infectious serositis, or duck septicemia, and clinical characteristics include diarrhea, lethargy, ocular and nasal discharge, respiratory symptoms (i.e., dyspnea, mild coughing and sneezing), and nervous symptoms (i.e., tremors of the head and neck or head and legs and ataxia) in addition to pathologic lesions such as fibrinous pericarditis, perihepatitis, airsacculitis, and meningitis [7,8,9,10].

To date, 21 RA serotypes with differences in virulence, both between and sometimes within a given serotype, have been identified, with less or no significant cross-protection among serotypes [11]. RA outbreaks have been documented in various countries, with the most commonly identified pathogenic isolates reported belonging to serotypes 1, 2, 3, 4, 5, and 15 [4,8,9,12,13,14]. Treatment for RA infection is carried out using antibiotics, but recent reports have shown drug resistance to various antibiotics, highlighting the need to identify novel therapeutic strategies [12,14]. Despite its prevalence, pathogenicity, lack of cross-protection among serotypes, and emerging antibiotic resistance, there is still a lack of understanding of the host–pathogen interaction, particularly in the context of the host immune response, during RA infection.

In developing new treatment strategies, understanding the immunologic pathways and nature of the immune response during a disease process is paramount to identifying immune targets and designing novel immune therapeutics. We have previously demonstrated the variable susceptibility of mice, chickens, and ducks to RA infection and the role of IL-17A in its pathogenesis [10,15]. To a lesser extent than IL-17A, related IL-17 cytokines and IL-17F, but not IL-17D, were also found to be upregulated in RA-infected ducks [15,16]. A comparative analysis of RA-infected chickens and ducks correlated with a more Th1/Th2-driven response in chickens, whereas a Th17-driven response was observed in ducks [15,17]. A targeted approach to reduce IL-17A and related Th17 cytokine expression was also attempted by our research group using natural substances (i.e., berberine and diindolylmethane) and recombinant IL-4 protein. The results showed that the treatment ameliorated the deleterious effects of RA infection, corroborating the findings on the involvement of IL-17 in the disease [10].

IL-17A is the signature cytokine of the IL-17 cytokine family, mainly produced by CD4^+^ T (Th17) cells, and has not only been implicated in humans for host protective immunity against various pathogens but also in tissue inflammation and autoimmune diseases [18,19]. Th17 cell differentiation is influenced by its cytokine environment, and one of the critical regulators for Th17 cell differentiation is IL-23 [20]. To understand the involvement of IL-17 regulation during RA infection in ducks, we analyzed related Th17 cytokines, including IL-23 and IL-22, and found that in RA infection, IL-23 does not play a critical role in the IL-17A response. Furthermore, in contrast to IL-22, IL-17A is produced by both CD4^+^ and other splenocytes [17,21]. Interestingly, we found an association of the common cytokine receptor γ chain with the expression of Th17-related cytokines during RA infection [22].

Although the participation of immune-related genes in RA infection has been studied, there is still a considerable knowledge gap in the understanding of the exact signaling and cellular network mechanisms underlying the host immune response during infection. Moreover, much progress and research are needed to elucidate the avian immune system in general, as it is still poorly understood, particularly in ducks, as evidenced by their limited gene annotations. In the present study, we performed RNA-seq of spleens collected on day four after infection based on histopathological examination to determine the host immune response to RA infection in ducks and to elucidate gene expression patterns and the potential regulatory and gene expression networks.

## 2. Materials and Methods

### 2.1. Animals and Ethics Statement

Pekin ducklings were obtained from Joowon ASTA Ducks (Changwon, Republic of Korea) and housed in battery cages with free access to feed and water within a temperature-controlled environment. Ducks were sacrificed at indicated time points in the experiment using atlanto-occipital dislocation. All procedures involving ducks were conducted in accordance with the Gyeongsang National University Guidelines for the Care and Use of Experimental Animals and were approved by the Institutional Animal Care and Use Committee (GNU-221011-E0132).

### 2.2. Animal Infection

RA serotype 1 was used for the infection protocol. The bacteria used in this study was isolated from a commercial duck farm in Changwon, Gyeongnam Province, Republic of Korea and was confirmed and serotyped at Chonbuk National University. Bacteria were initially cultured on blood agar plates containing 5% sheep blood (Asan Pharmaceutical, Seoul, Republic of Korea) for 48 h in an incubator at 37 °C with 5% CO_2_. Subsequently, a single colony was selected, inoculated in tryptic soy broth (Difco, Tucker, GA, USA), and cultured in a shaking incubator at 37 °C with vigorous shaking until reaching the exponential growth phase as previously described [15]. Ten-fold serial dilutions of the bacterial stock were plated on 5% sheep blood agar plates to determine the final concentration of the stock to be used as the inoculum. Twenty two-week-old ducks were injected via the intramuscular route in the thigh using a standard needle (26 gauge) with 500 CFU of RA serotype 1 (RA-infected group, dRA, n = 20) suspended in 200 μL of phosphate-buffered saline (PBS). A similar procedure was followed for the uninfected negative control group (dNC, n = 20), with an injection of 200 μL of PBS alone. At 4- and 7-days post-infection (dpi), spleens (n = 5) were collected from each group for histopathological examination. Additionally, spleens were collected from three ducks per group at 4 dpi for subsequent gene expression analysis by RNA-seq.

### 2.3. Histopathological Examination

The spleen tissues from each group were fixed in 10% neutral-buffered formalin and routinely processed by dehydration and paraffin embedding for subsequent histopathological examination. A 4 µm tissue section of each sample was stained with hematoxylin and eosin (H&E) and examined under an optical microscope. The tissue sections were evaluated for the presence of lesions.

### 2.4. Library Construction and Transcriptome Sequencing

RNA-seq library construction was conducted using TruSeq DNA PCR, and sequencing was performed on an Illumina HiSeq4000 platform (eGnome, Seongnam, Republic of Korea). FastQC (v0.11.8) was utilized to filter Fast-Q formatted raw reads to obtain high-quality reads. Trimmomatic (v0.39) was applied to trim adapter sequences with quality scores less than 20 from the raw reads. The resulting high-quality reads were mapped to the *Anas platyrhynchos* (CAU_duck1.0; GCA_002743455.1) reference genome using HISTAT2 (v2.2.1), and FeatureCount (v2.0.2) was employed to quantify reads in terms of the genomic features.

### 2.5. Transcriptome Data Analysis

Normalization and differential expression analyses were conducted with edgeR (v3.32.1) using the trimmed mean of M (TMM) values. The distribution, distance, and overall relationship of the samples and TMM value clustering of the expression patterns between samples were visualized using multidimensional scaling plots and heat maps, respectively. The average values of three samples from the dNC group and three samples from the dRA group were used for all downstream analyses for each group. The *p*-values were adjusted using the Benjamini–Hochberg method for controlling the false discovery rate (FDR), and significantly differentially expressed genes (DEGs) were defined as genes with a *p*-adjusted FDR < 0.05 and Log_2_ |fold change| (Log_2_FC) ≥ 1 [23]. Due to the limited annotations of duck genes, functional clustering and pathway enrichment analysis of DEGs were performed indirectly as described elsewhere, with modifications [24]. First, the duck ensemble gene IDs were converted to their chicken ortholog identifiers using the g:Orth function in g:Profiler (version: e111_eg58_p18_30541362, https://biit.cs.ut.ee/gprofiler/orth, accessed on 19 February 2024), and then the matched gene IDs were used for subsequent analyses. The matched gene ID was used in g:GOSt in g:Profiler (version: e111_eg58_p18_30541362, https://biit.cs.ut.ee/gprofiler/gost, accessed on 19 February 2024), with *Homo sapiens* as the input organism and an FDR threshold < 0.05 for the functional analysis of the DEGs using the Gene Ontology (GO) terms for the biological process (BP), molecular function (MF), and cellular component (CC) domains. Additionally, Kyoto Encyclopedia of Genes and Genomes (KEGG) pathway enrichment analysis was performed. Our previous reports indicated differential expression of genes in ducks and chickens during RA infection; hence, *Homo sapiens* was used instead of *Gallus gallus* for the enrichment analysis [15,17]. A search tool for retrieving interacting genes/proteins (STRING v12.0, https://string-db.org/, accessed on 5 March 2024) was used for protein–protein interaction (PPI) network analysis of DEGs. Unless otherwise stated, all graphs were created using the ggplot2 package in R version 4.3.2 for all analyses.

## 3. Results

### 3.1. RA Infection Caused Lymphoid Depletion in the Duck Spleen

RA serotype 1 is considered a pathogenic RA serotype [4]. In our previous report, we found that following RA serotype 1 infection in ducks, the survival rate was 20%, and the bacterial burden was highest in the liver and spleen of ducks on day 4 [15]. Inoculation with either RA serotype 1 or serotype 7 also resulted in greater upregulation of IL-17A and IL-17F expression in the spleen and liver of infected ducks at 4 dpi than at 7 dpi compared to their respective uninfected controls [15]. To determine the pathological impact of RA infection on the tissues, a histopathological examination of the spleen and liver was performed at 4 and 7 dpi (Figure 1 and Appendix A). The spleens of RA-infected ducks generally exhibited marked lymphoid depletion at 4 dpi rather than at 7 dpi. Additionally, the boundary between the white pulp and red pulp in the infected ducks was less pronounced at both time points. More specifically, the infected ducks developed a more diffuse white pulp along with a reduced lymphoid follicle structure (Figure 1). Compared with the spleen, the liver of RA-infected ducks showed marked congestion, steatosis, and inflammation (Appendix A). Collectively, the histological findings from this study and the aforementioned reports indicated that the spleen samples collected at 4 dpi represented the peak of infection and were therefore selected for analysis of the mechanisms involved in the prominent expression of IL-17A in subsequent studies.

### 3.2. Distinct Transcriptome Profile in the Spleens of RA-Infected Ducks

Duck spleen samples collected at 4 dpi were subjected to gene expression analysis using RNA-seq to elucidate the gene expression profile and host immune responses. The sequencing generated 19,657.5 M bases and 194.4 M raw reads from six duck spleen samples. Uninfected control samples from ducks (dNC) contained an average of 30.6 ± 0.5 M raw reads. RA-infected samples (dRA) contained an average of 34.2 ± 2.8 M raw reads. After removing adapter reads and low-quality reads, a total of 161.6 M clean reads and an average of 26.9 ± 1.1 M clean reads were obtained, corresponding to 83.1 ± 1.2% of raw reads. The average GC content and Q30 of these clean reads were 48.5 ± 0.3% and 96.2 ± 0.2%, respectively (Appendix A).

A multidimensional scaling plot and hierarchical clustering heatmap comprising the 500 most variable genes exhibited a distinct separation between the dRA and dNC groups, indicating an altered transcriptomic profile in the spleens of infected ducks compared with those of uninfected ducks (Figure 2A,B). The individually computed Log_2_FC was used to represent the total DEGs. Among the initial 9272 genes identified through mRNA sequencing, 287 genes were significantly differentially expressed between the spleens of infected and uninfected ducks (Figure 2C). Of these DEGs, 78% (224 genes) were upregulated, and 22% (63 genes) were downregulated. Additionally, the DEGs included 53 (23.66%) and 10 (15.87%) novel genes that were upregulated and downregulated, respectively (Table 1). A complete list of the novel DEGs is shown in Appendix A.

### 3.3. The Top Significantly Regulated DEGs Included Cytokines, Enzymes, and Receptors

Among the DEGs, the top upregulated and downregulated genes identified are shown in Table 2 and Table 3, respectively. The top upregulated DEGs included cytokines (IL-17A, CCL15, IL8L1, IL26, CCLL4, CCL4, and IL-17F), enzymes (TRIM35, APOBEC1, VNN2, ACOD1, MMP10, NOS2, and K123), transporters (TMPRSS2 and EXFABP), mucin (MUC2), a receptor protein (MRGPRH), membrane-anchored glycoprotein (MSLNL), and a hormone (STC2) (Table 2). As shown in Table 3, the top downregulated DEGs included a hormone (GRP), an enzyme (GDPD2), a transporter (PKD2L1), stimulator or activator proteins (SCRG1 and ARHGAP20), transmembrane proteins (PCDH18, TPBG, and TMEM86A), a voltage-gated calcium channel (CACNG5), a plasma protein (HBE1), refilin (RFLNA), receptors (CHRNA3, RTN4RL1, PGR, and CD36), a metabolic protein (FABP5), regulatory proteins (NREP and KLHDC8B), a developmental protein (NTNG2), and adipokine (C1QTNF4).

### 3.4. GO Classification and Function Analysis of DEGs Showed Significant Association with the Response to External Stimuli and Cytokine-Mediated Signaling Pathways for Upregulated DEGs and Extracellular Matrix and Collagen Trimer for Downregulated DEGs

Enrichment analysis of the DEGs was carried out using a GO enrichment analysis tool for functional profiling and characterization of the gene set and its products based on the MF, BP, and CC domains. The results were interpreted based on the degree of the negative base 10 logarithm of the adjusted *p*-value (−log10 *p*-value) and fold enrichment. The enrichment analysis of upregulated DEGs showed a total of 767 enriched GO terms with 28 in the MF, 718 in the BP, and 21 in the CC domains. The 15 highest-ranked significantly enriched GO terms for each group are shown in Figure 3. In the MF domain, the most significant terms were cytokine receptor activity (−log10 *p*-value: 5.35), immune receptor activity (−log10 *p*-value: 4.32), cytokine binding (−log10 *p*-value: 4.32), and hydrolase activity (−log10 *p*-value: 3.8). The highest fold enrichment of genes with regard to associated terms included NAD+ nucleoside activity (fold enrichment: 34.67), amine binding (fold enrichment: 25.42), NADP+ nucleosidase activity (fold enrichment: 23.83), and mitogen-activated protein kinase binding (fold enrichment: 18.16) (Figure 3A).

The majority of the enriched terms among the upregulated DEGs were associated with the BP domain, and the highest-ranked significant terms were response to external stimulus (−log10 *p*-value: 21.44), defense response (−log10 *p*-value: 20.56), and response to other organism and external biotic stimulus (−log10 *p*-value: 19.70). However, overrepresentation of genes was observed for the following terms: cytokine-mediated signaling pathway (fold enrichment: 34.799), cellular response to cytokine stimulus (fold enrichment: 6.22), inflammatory response (fold enrichment: 6.13), and response to cytokine (fold enrichment: 6.05) (Figure 3C). Finally, the significantly enriched GO terms for the CC domain included extracellular region (−log10 *p*-value: 8.38), extracellular space (−log10 *p*-value: 5.70), cell periphery (−log10 *p*-value: 3.57), and cytoplasm (−log10 *p*-value: 3.49). Additionally, the highest fold enrichment of the genes in the gene set was found for germ plasm (fold enrichment: 16.14), cytoplasmic ribonucleoprotein granule (fold enrichment: 4.39), external side of plasma membrane (fold enrichment: 4.37), and side of membrane (fold enrichment: 3.10) (Figure 3E). Accordingly, all associated genes for each term along with their related Log_2_FC are listed in Figure 3B,D,F. Moreover, IL17A was associated with the majority of the highest-ranked significantly enriched GO terms. Similar to IL17A, CCL19, CCL4, CCL5 and IL17F were associated with all of the top 15 highest-ranked GO terms in the BP domain. Interestingly, ACOD1, ASS1, CLDN1, and NOS2 were also associated with the majority of the 15 highest-ranked BP domain terms (Figure 3D).

GO analysis of the downregulated DEGs identified a total of 70 enriched GO terms with 42 in the BP and 28 in the CC domain. There were no significantly enriched terms found for the MF domain. As above, the top 15 highest-ranked statistically significant terms along with the highest-ranked related genes from the gene set for each term are listed (Appendix A). In the BP domain, the highest statistically significant terms were nervous system development (−log10 *p*-value: 2.49), multicellular organismal process (−log10 *p*-value: 2.22), and neuropeptide catabolic process (−log10 *p*-value: 1.92). However, the fold enrichment was less than one, indicating that the biological relevance of the gene set to the BP terms is underrepresented compared with that of the dataset in the database (Appendix A). The highest-ranked CC domain GO terms of the downregulated DEGs were extracellular matrix (−log10 *p*-value: 4.37), external encapsulating structure (−log10 *p*-value: 4.37), extracellular space (−log10 *p*-value: 4.23), and cell periphery (−log10 *p*-value: 3.84). Additionally, the fold enrichment was highest for collagen trimer (fold enrichment: 14.57), Schaffer collateral-CA1 synapse (fold enrichment: 13.53), postsynaptic density membrane (fold enrichment: 13.13), and extracellular matrix (fold enrichment: 7.96) (Appendix A). GO analysis highlighted the association of CD36, FABP5, LRRC4C, SLC16A7, MME, and MMP2 with the significantly enriched GO terms in the BP and CC domains among downregulated DEGs (Appendix A).

### 3.5. KEGG Pathway Enrichment Analysis of DEGs Identified Significant Cytokine–Cytokine Receptor Interactions

To further understand the related pathway maps of the DEGs, KEGG enrichment analysis was performed on the gene set (Figure 4). Similarly, the enriched term results were interpreted based on their significance according to the −log10 adjusted *p*-value and the rich factor. KEGG enrichment analysis showed a total of 15 enriched pathways, with the top significantly enriched pathway being the cytokine–cytokine receptor interaction (−log10 *p*-value: 7.35), followed by the JAK-STAT signaling pathway (−log10 *p*-value: 3.39) and viral protein interaction with cytokine and cytokine receptors, osteoclast differentiation, and influenza A (−log10 *p*-value: 2.86). Additionally, the proportion of the genes present from the gene set in relation to the genes of a particular KEGG pathway was high for leishmania (rich factor: 0.083), viral protein interaction with cytokine and cytokine receptors (rich factor: 0.082), complement and coagulation cascades (rich factor: 0.081), and cytokine–cytokine receptor interaction (rich factor: 0.069) (Figure 4A). A list of associated genes for each enriched pathway with their related expression is shown in Figure 4B. Specifically, upregulated DEGs involved in cytokine–cytokine receptor interactions included IL17A, CCL15, IL26, CCL4, IL17F, CCL19, IL10, CCL5, ACKR4, IL5RA, IL2RA, TNFRSF11B, IL1RAP, ACVR2A, IL13RA2, TNFRSF11A, IL31RA, IFNGR2, CXCR4, and IL1R1. Interestingly, KEGG analysis further showed pathway enrichment for Th17 cell differentiation (−log10 *p*-value: 2.28, rich factor: 0.067) for the related genes IL17A, IL17F, IL2RA, IL1RAP, IFNGR2, IL1R1, and NFKBIA among the upregulated DEGs (Figure 4B). KEGG analysis of the downregulated DEGs only resulted in one significantly enriched pathway, the PPAR signaling pathway (−log10 *p*-value: 1.38, rich factor: 0.04), involving the genes CD36, FABP5, and FADS2 (Appendix A).

### 3.6. PPI Network Analysis of DEGs Showed Clustering of Proteins Involved in Cytokine–Cytokine Interactions, the Inflammatory Response, Th17 Cell Differentiation, and the IL-17 Signaling Pathway

PPI network analysis of DEGs with a minimum interaction score of 0.7 showed close clustering of proteins associated with cytokine–cytokine interactions, the inflammatory response, Th17 cell differentiation, and the IL-17 signaling pathway. Genes associated with the PPAR signaling pathway formed a separate cluster but interacted with VCAM1, which was similarly shown to be related to the NF-kappa B and TNF signaling pathways. Interestingly, the PPI network also highlighted the high level of interaction between IL-17A and IL-17F, as well as the interactions of IL17A with IL-10, IL26, NFKBIA, NFKB1, IL1R1, and IL2RA (Figure 5).

### 3.7. RNA-Seq Gene Lists Included Many Expressed Genes Related to Th17 Differentiation and IL-17 Signaling

All enrichment and PPI networks of DEG analyses indicated a high degree of cytokine-mediated signaling and the association of IL17 with the majority of the results from these analyses. Hence, to further elucidate these findings, a complete list of genes related to Th17 cell differentiation and the IL-17 signaling pathway was identified not only among DEGs but also for the RNA-seq results. In the KEGG database, 105 genes were indicated as related to Th17 cell differentiation (ID: KEGG04659). In the KEGG gene list, 30 upregulated (28.57%) and 15 (14.28%) downregulated genes were present in the spleens of RA-infected ducks (Table 4). More specifically, seven upregulated genes were DEGs. Therefore, 42.85% (45/105) of the genes associated with Th17 differentiation was expressed in RA-infected samples (Table 4). Conversely, 92 genes were related to IL17 signaling (ID: KEGG:04657) in the KEGG database. According to the RNA-seq results, 22 (23.91%) and 15 (16.30%) genes were upregulated and downregulated, respectively (Table 5). Of the upregulated genes, four were DEGs. Accordingly, a total of 40.21% (37/92) was expressed during RA infection (Table 5). The related downregulated genes were not differentially expressed compared with those in uninfected controls.

## 4. Discussion

Host–pathogen interactions are complex processes involving the host and pathogen in all stages of disease pathogenesis. Although hosts have developed defense mechanisms against diverse microbial pathogens, pathogens have also evolved and developed distinct strategies to evade the host response to cause disease [25]. Hence, understanding and investigating the interplay between hosts and pathogens is crucial for the prevention and treatment of disease and for the development of novel therapeutic approaches. Activation of the host’s innate immune response during infection results in the production of multiple effector molecules such as cytokines, chemokines, and antimicrobial peptides against the pathogen [26,27,28]. However, dysregulation of these effector molecules, especially cytokines, can lead to immunopathology instead of homeostasis, such as in the case of autoimmune and inflammatory diseases [18,19]. RA infection causes high mortality and morbidity rates in ducks, and previous studies have indicated a role of IL-17A in the pathogenesis of infection. However, the signaling and cellular networks related to these findings are still lacking and poorly understood. RNA-seq is a powerful, robust, and adaptable technique to measure the complex expression dynamics of genes in various physiological states at the genome-wide level. Therefore, the gene expression pattern in RA-infected ducks was investigated using RNA-seq to determine and identify key regulators and potential targets applicable to the development of innovative therapeutic strategies.

RA infection can cause gross lesions and a spotted spleen with mild pericarditis, liver capsule hepatorrhagia, and cerebral hemorrhage in Muscovy ducks. The infected ducks histo-pathologically develop extensive white myeloid fibrinoid necrosis with a marked loss of cell structure or residual necrotic fragmented nuclei, an unclear boundary between red and white pulp, more diffuse white pulp, and no obvious inflammatory infiltrates in the spleen [9]. Our study showed similar findings, suggesting that RA infection impairs and disorganizes the spleen compartments, affecting the immunological function of the spleen because it is the site of homing and differentiation of immune cells (i.e., macrophages, monocytes, granulocytes, dendritic cells, and plasma cells) [29]. Additionally, the spleens of infected ducks showed marked lymphoid depletion, indicating apoptosis and necrosis. Accordingly, the GO enrichment analysis of DEGs in the BP domain indicated a total of 34 enriched terms associated with leukocyte migration, differentiation, activation, and regulation (Appendix A). Interestingly, IL-17A was also related to the terms leukocyte, myeloid leukocyte differentiation, and myeloid leukocyte migration, but not leukocyte apoptosis (Appendix A). The analysis of upregulated DEGs identified IL10 and CD274 as associated with positive regulation of leukocyte apoptosis, suggesting their involvement in the apoptotic process of RA infection. Expression of IL10 and CD274 (also known as programmed death ligand 1) was associated with tumors, such as in the clinical manifestation of laryngeal squamous cell carcinoma and colorectal cancer-derived liver metastases [30,31,32].

In the spleen, the organization of the white pulp and the marginal zone is controlled by lipid mediators and adhesion molecules as well as chemokines [29]. PPARs are nuclear hormone receptors that function in lipid metabolism and metabolic control [33,34]. In this study, the PPAR signaling pathway was significantly enriched among downregulated DEGs, suggesting the association of this signaling pathway with the histopathological lesions observed in the spleen. Interestingly, the PPI network analysis showed that PPAR signaling interacts with VCAM1. VCAM1 was significantly upregulated in RA infection, indicating its role in the downregulation of PPAR signaling. An inverse relationship between PPARs and the VCAM1 response was also found in PPAR activator-treated endothelial cells [35]. Although spleens were preferentially used in this study because the whole organ can be used for analysis, time-matched histopathological sections of the liver were also examined. In contrast to the spleen, the livers of RA-infected ducks showed marked congestion, steatosis, and inflammation, warranting a potential differential host immune response that will require further study (Appendix A).

In accordance with previous reports, RNA-seq results unequivocally demonstrated significantly upregulated expression of both IL-17A and IL-17F in RA infection. Additionally, all enrichment analyses underscored the involvement of IL-17A in the majority of functional analyses, and the highly enriched cytokine-mediated response in RA infection in ducks involved a network of interactions among IL-17A, IL-17F, CCL15, IL8L1, IL-26, CCLL4, CCL4, CCL19, IL-10, CCL5, ACK-R4, IL-5RA, IL2RA, TNFRSF-11B, IL1-RAP, ACV-R2A, IL13-RA2, TNFRSF-11A, IL31-RA, IFNG-R2, CXC-R4, and IL1-R1. Similarly, we have previously demonstrated the association of the common receptor γ chain, part of IL-2RA, in the expression of Th17-related cytokines during RA infection [22]. In the present study, in addition to IL-2RA, we were able to identify new potential interactions of IL17A that may be involved in the disease process of RA, including interactions with IL-10, IL26, NFKBIA, NFKB1, and IL1R1.

IL-10 is a Th2 cytokine that participates in regulating IL-17-mediated pathology [36]. A recent report indicated the pivotal role of IL-26 in bridging the Th17 and Th2 immune response in the development of atopic dermatitis [37]. IL-26 is a member of the IL-10 family, which has diverse antimicrobial functions and has been implicated in the pathology of lymphatic filariasis, atopic dermatitis, and Crohn’s disease, as well as in mediating Th17-associated diseases such as rheumatoid arthritis and psoriasis [37,38,39,40,41]. Moreover, IL-26 is an emerging upstream novel proinflammatory cytokine that may induce Th17 cell generation and the proinflammatory cascade [40,42,43]. IL-26 in this study was also found to be significantly differentially upregulated in RA infection. In the PPI network analysis of all DEGs, a positive relationship or interaction was identified between IL26 and IL17A (Figure 5), indicating the possibility that RA infection stimulates IL-17A expression with involvement of IL-26. Additional clarification and investigations are needed to verify this finding.

## 5. Conclusions

We have previously found that regulation of IL-17A in RA infection is not linked to IL-23 [21]. RNA-seq analysis was used to investigate other mechanisms of IL-17A upregulation in RA infection. We identified a possible interaction of IL-26 with IL-17, highlighting its potential as a novel upstream cytokine that can regulate IL-17 in RA infection. Additionally, this process identified several important pathways and genes related to the complex networks and potential regulation of the host immune response in RA-infected ducks. Collectively, these findings could not only serve as a roadmap in our understanding of RA infection and the development of novel immunotherapeutic approaches against the disease, but also provide an opportunity to understand the avian immune system.

## Figures and Tables

**Figure 1 microorganisms-12-00973-f001:**
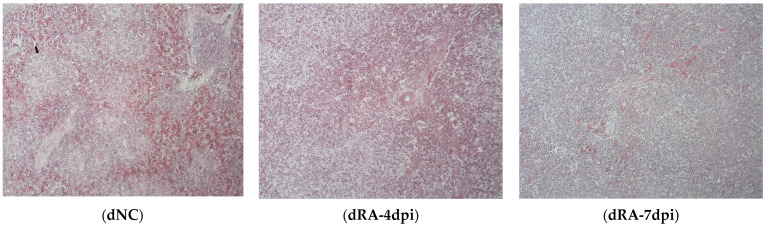
Histopathologic lesions in the spleens of *Riemerella anatipestifer* (RA)–infected ducks. Two-week-old ducks were intramuscularly inoculated with RA (5 × 10^2^ CFU), and spleens were collected at 4 and 7 days post-infection (dpi). Spleen samples were sectioned and stained with hematoxylin and eosin (H&E) for histopathological examination. The spleens of RA-infected ducks (dRA) showed decreases in the number of lymphocytes and structure of the white pulp (arrow), including the lymphoid follicles. Images are shown at 100× magnification and are representative of the five ducks in each group. dNC, uninfected negative control.

**Figure 2 microorganisms-12-00973-f002:**
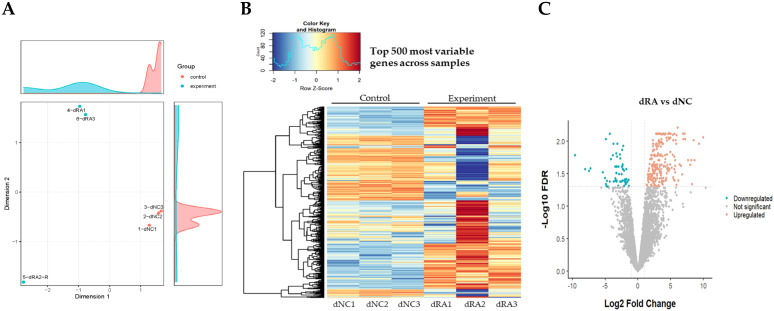
Gene expression patterns in uninfected negative control (dNC) versus RA-infected ducks (dRA). (**A**) Multidimensional scaling plot depicting sample distribution of RNA-sequencing (RNA-seq) datasets. Each point represents an individual sample, with the distance between two points calculated as their leading fold change (FC). (**B**) Heatmap and dendrogram visualizing expression patterns between samples through clustering trimmed mean of M values of the top 500 most variably expressed genes. Each row and column represent a sample and a gene, respectively. Significant differentially expressed genes (DEGs) were filtered according to a Log_2_ |FC| value ≥ 1. Red represents upregulated DEGs, and blue represents downregulated DEGs relative to the mean expression of all six samples. (**C**) Volcano plot of the DEGs in RA-infected ducks. Each dot represents a gene. Orange and blue dots indicate upregulated and downregulated DEGs, respectively, and gray dots indicate non-differentially expressed genes. The dotted lines represent the FC and false discovery rate (FDR) cut-off values of the differential expression.

**Figure 3 microorganisms-12-00973-f003:**
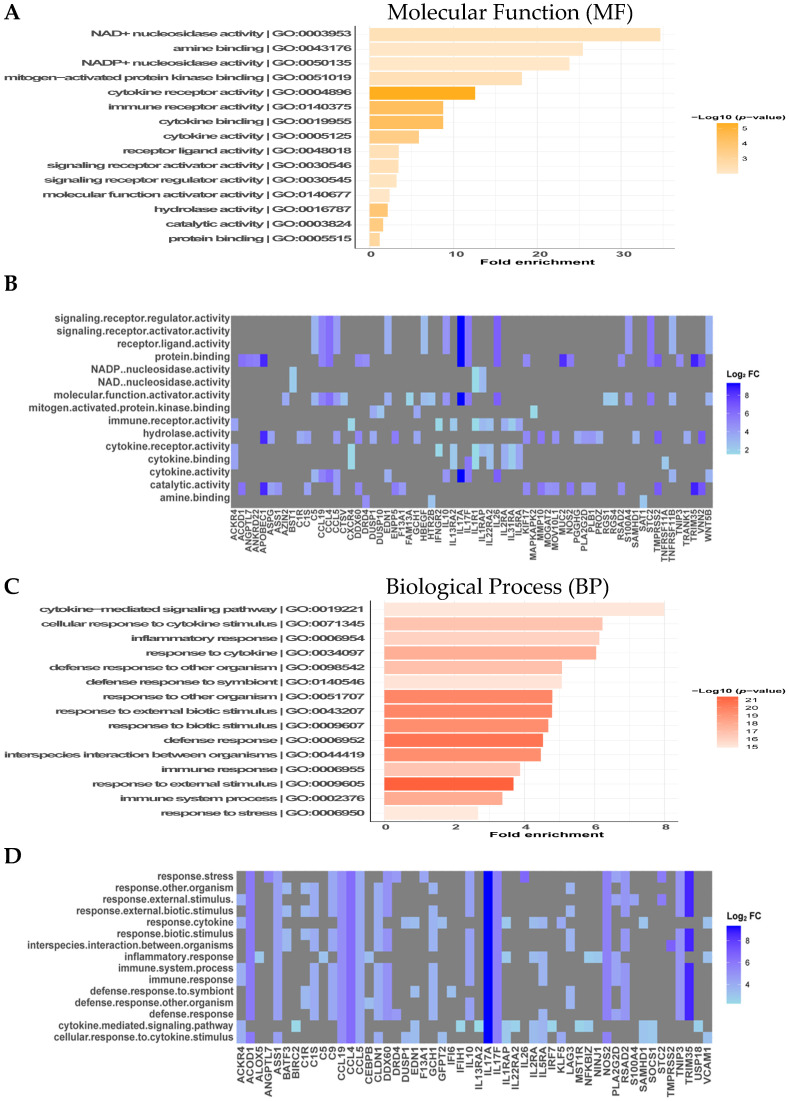
Gene Ontology (GO) enrichment analysis of upregulated DEGs in ducks infected with RA. Bar graphs show the top 15 highest-ranked significantly enriched GO terms for the (**A**) molecular function (MF), (**C**) biological process (BP), and (**E**) cellular component (CC) domains. The statistical significance of each enriched GO term in the gene set is indicated as the negative base 10 logarithm of the adjusted *p*-value (−log10 *p*-value), and overrepresentation of genes associated with a term in the gene set is shown as the fold enrichment. A high −log10 *p*-value and fold enrichment indicate that the term is both statistically significant and biologically relevant. GO term IDs are indicated for each term name, and the color of the scale indicates the relationship of the −log10 *p*-value for each term. Heatmaps of the related genes in the (**B**) MF, (**D**) BP, and (**F**) CC domains are also shown and are colored based on their Log_2_FC according to their expression in the RNA-seq data set. Gray indicates that the gene is not associated with the term.

**Figure 4 microorganisms-12-00973-f004:**
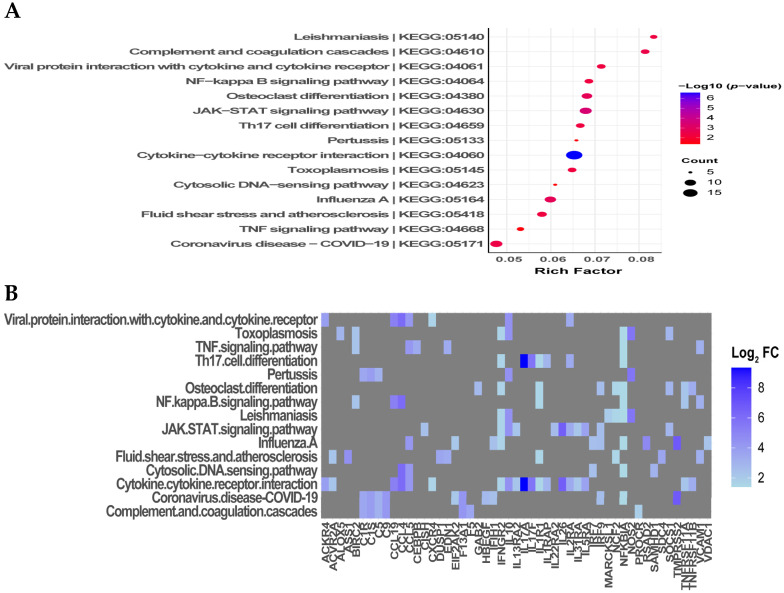
Kyoto Encyclopedia of Genes and Genomes (KEGG) pathway enrichment analysis of upregulated DEGs in RA-infected ducks. (**A**) Enriched KEGG pathway terms of the upregulated DEGs. The rich factor of each top enriched pathway is represented by the *x*-axis. The rich factor represents the ratio of the number of DEGs to the number of genes present for the specific pathway term. The size and color of the dots represent the number of enriched genes and the significance of each pathway, respectively. KEGG term IDs are indicated for each term name. (**B**) List of genes related to each enriched pathway. A heatmap shows the expression level of related DEGs for each term. Gray indicates that the gene is not associated with the term.

**Figure 5 microorganisms-12-00973-f005:**
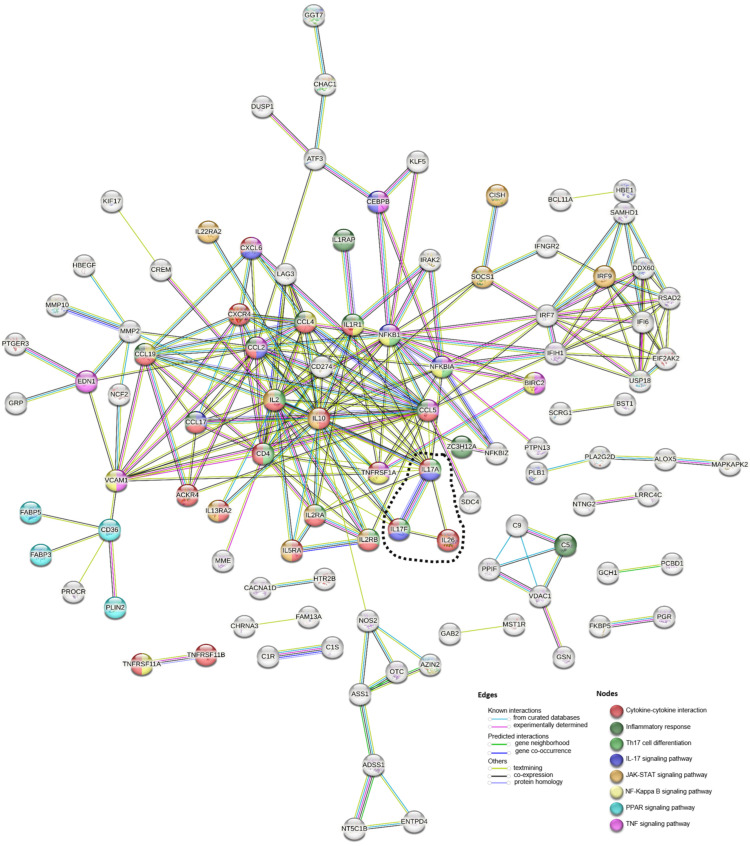
Protein–protein gene network in RA-infected ducks. An interaction analysis of all DEGs was performed using STRING version 12 (http://string-db.org/ accessed on 5 March 2024). Each line and node correspond to interactions and significantly enriched related pathways. The interaction of IL-26 and IL-17 is highlighted in dotted line. The network was inferred using a high confidence (0.700) interaction score using the full STRING network. Disconnected nodes were not included.

**Table 1 microorganisms-12-00973-t001:** List of differentially expressed genes (DEGs) at 4 days post-infection in spleens of *Riemerella anatipestifer*-infected ducks.

Number of Genes Qualifying the Criteria	Total Genes			Total Upregulated DEGs			Total Downregulated DEGs		
	Differentially Expressed	Upregulated	Downregulated	Differentially Expressed	Known Genes	Novel Genes	Differentially Expressed	Known Genes	Novel Genes
9272	287	224	63	224	171	53	63	53	10

Significant DEGs were filtered by Log2 |Fold change| ≥ 1, *p*-adj false discovery rate < 0.05.

**Table 2 microorganisms-12-00973-t002:** The top 20 upregulated DEGs in the spleens of *Riemerella anatipestifer*-infected ducks.

Gene ID	Gene Name	Log_2_FC
IL17A	Interleukin 17A	9.34
TRIM35	Tripartite motif containing 35	8.65
APOBEC1	Apolipoprotein B mRNA editing enzyme catalytic subunit 1	8.51
MUC2	Mucin 2	8.28
CCL15	C-C motif chemokine ligand 15	8.18
MRGPRH	MAS-related G protein-coupled receptor H	7.23
VNN2	Vanin 2	6.95
IL8L1	Interleukin 8-like 1	6.87
TMPRSS2	Transmembrane serine protease 2	6.78
IL26	Interleukin 26	6.58
CCLL4	Chemokine-like ligand 4	6.37
MSLNL	Mesothelin-like	6.14
CCL4	C-C motif chemokine ligand 4	6.07
ACOD1	Aconitate decarboxylase 1	5.92
STC2	Stanniocalcin 2	5.84
EXFABP	Extracellular fatty acid-binding protein	5.76
MMP10	Matrix metallopeptidase 10	5.69
IL17F	Interleukin 17F	5.68
NOS2	Nitric oxide synthase 2	5.61
K123	Endonuclease domain containing 1	5.60

FC = Fold change.

**Table 3 microorganisms-12-00973-t003:** The top 20 downregulated DEGs in the spleens of *Riemerella anatipestifer*-infected ducks.

Gene ID	Gene Name	Log_2_FC
GRP	Gastrin releasing peptide	−9.64
GDPD2	Glycerophosphodiester phosphodiesterase domain containing 2	−8.07
PKD2L1	Polycystin 2 like 1, transient receptor potential cation channel	−7.49
SCRG1	Stimulator of chondrogenesis 1	−5.39
PCDH18	Protocadherin 18	−5.04
CACNG5	Calcium voltage-gated channel auxiliary subunit gamma 5	−4.91
TPBG	Trophoblast glycoprotein	−4.90
HBE1	Hemoglobin subunit epsilon 1	−4.72
RFLNA	Refilin A	−4.62
CHRNA3	Cholinergic receptor nicotinic alpha 3 subunit	−4.29
FABP5	Fatty acid binding protein 5	−4.15
NREP	Neuronal regeneration related protein	−3.71
RTN4RL1	Reticulon 4 receptor like 1	−3.50
TMEM86A	Transmembrane protein 86A	−3.47
KLHDC8B	Kelch domain containing 8B	−3.45
ARHGAP20	Rho GTPase activating protein 20	−3.33
NTNG2	Netrin G2	−3.31
C1QTNF4	C1q and TNF related 4	−3.29
PGR	Progesterone receptor	−3.28
CD36	CD36 molecule	−3.26

FC = Fold change.

**Table 4 microorganisms-12-00973-t004:** List of genes found in the spleens of *Riemerella anatipestifer*-infected ducks related to Th17 differentiation.

Expression	Gene	Description	Log_2_FC	FDR
DEG				
	IL17A	Interleukin-17A	9.34	0.01
	IL17F	Interleukin-17F	5.68	0.02
	IL2RA	Interleukin-2 receptor subunit alpha	3.38	0.02
	IL1RAP	Interleukin 1 receptor accessory protein	2.90	0.01
	IFNGR2	Interferon gamma receptor 2	1.63	0.04
	IL1R1	Interleukin 1 receptor type 1	1.54	0.04
	NFKBIA	NF-kappa-B inhibitor alpha	1.40	0.05
Upregulated				
	GATA3	GATA binding protein 3	2.30	0.33
	IFNG	Interferon gamma	2.30	0.18
	CHUK	Component of inhibitor of nuclear factor kappa B kinase complex	2.12	0.18
	RORA	RAR related orphan receptor A	1.65	0.39
	HIF1A	Hypoxia inducible factor 1 subunit alpha	1.62	0.23
	JUN	Transcription factor AP-1	1.56	0.13
	STAT1	Signal transducer and activator of transcription 1	1.45	0.15
	NFKBIE	NF-kappa-B inhibitor epsilon	1.41	0.09
	STAT3	Signal transducer and activator of transcription 3	1.37	0.11
	IL6ST	Interleukin-6 receptor subunit beta	1.09	0.08
	IL21R	Interleukin-21 receptor	1.08	0.34
	RXRA	Retinoic acid receptor RXR-alpha	1.04	0.31
	FOS	Proto-oncogene c-Fos	0.93	0.29
	IL2RB	Interleukin-2 receptor subunit beta	0.93	0.53
	IL6R	Interleukin-6 receptor subunit alpha	0.76	0.46
	MAPK11	Mitogen-activated protein kinase 11	0.51	0.52
	IL21	Interleukin-21	0.42	0.86
	MAPK14	Mitogen-activated protein kinase 14	0.41	0.52
	JAK2	Tyrosine-protein kinase JAK2	0.27	0.86
	PRKCQ	Protein kinase C theta type	0.27	0.77
	SMAD3	SMAD family member 3	0.16	0.82
	PLCG1	1-phosphatidylinositol 4,5-bisphosphate phosphodiesterase gamma-1	0.08	0.94
	ZAP70	Tyrosine-protein kinase ZAP-70	0.01	0.99
Downregulated				
	NFATC1	Nuclear factor of activated T-cells, cytoplasmic 1	−1.09	0.34
	MAPK9	Mitogen-activated protein kinase 9	−0.83	0.30
	LCK	Tyrosine-protein kinase Lck	−0.76	0.53
	CD3E	T-cell surface glycoprotein CD3 epsilon chain	−0.45	0.76
	CD247	T-cell surface glycoprotein CD3 zeta chain	−0.44	0.73
	PPP3CB	Serine/threonine-protein phosphatase 2B catalytic subunit beta isoform	−0.34	0.72
	TGFBR1	TGF-beta receptor type-1	−0.30	0.83
	IKBKB	Inhibitor of nuclear factor kappa-B kinase subunit beta	−0.28	0.72
	MTOR	Serine/threonine-protein kinase mTOR	−0.25	0.80
	JAK1	Tyrosine-protein kinase JAK1	−0.17	0.86
	RUNX1	Runt-related transcription factor 1	−0.12	0.91
	IFNGR1	Interferon gamma receptor 1	−0.08	0.96
	MAPK12	Mitogen-activated protein kinase 12	−0.07	0.94
	MAPK3	Mitogen-activated protein kinase 3	−0.07	0.94
	PPP3R1	Calcineurin subunit B type 1	−0.07	0.94

FC = Fold change; FDR = false discovery rate.

**Table 5 microorganisms-12-00973-t005:** List of genes found in the spleens of *Riemerella anatipestifer*-infected ducks related to the IL17 signaling pathway.

Expression	Gene	Description	Log_2_FC	FDR
DEGs				
	IL17A	Interleukin-17A	9.34	0.01
	IL17F	Interleukin-17F	5.68	0.02
	CEBPB	CCAAT/enhancer-binding protein beta	3.37	0.01
	NFKBIA	NF-kappa-B inhibitor alpha	1.40	0.05
Upregulated				
	IFNG	Interferon gamma	2.30	0.18
	CHUK	Component of inhibitor of nuclear factor kappa B kinase complex	2.12	0.18
	PTGS2	Prostaglandin G/H synthase 2	1.89	0.15
	TAB2	TGF-beta activated kinase 1 (MAP3K7) binding protein 2	1.67	0.39
	JUN	Transcription factor AP-1	1.56	0.13
	TRAF2	TNF receptor associated factor 2	1.38	0.06
	TRAF3IP2	Adapter protein CIKS	1.25	0.17
	GSK3B	Glycogen synthase kinase-3 beta	1.16	0.29
	FADD	FAS-associated death domain protein	1.08	0.19
	TNFAIP3	Tumor necrosis factor alpha-induced protein 3	1.04	0.46
	IKBKE	Inhibitor of nuclear factor kappa-B kinase subunit epsilon	0.98	0.18
	FOS	Proto-oncogene c-Fos	0.93	0.29
	MAPK6	Mitogen-activated protein kinase 6	0.79	0.19
	HSP90B1	heat shock protein 90 beta family member 1	0.70	0.46
	MAPK11	Mitogen-activated protein kinase 11	0.51	0.52
	CASP3	Caspase 3	0.43	0.81
	MAPK14	Mitogen-activated protein kinase 14	0.41	0.52
	TRAF6	TNF receptor associated factor 6	0.29	0.81
Downregulated				
	MAPK9	Mitogen-activated protein kinase 9	−0.83	0.30
	TRAF4	TNF receptor associated factor 4	−0.82	0.38
	MAPK4	Mitogen-activated protein kinase 4	−0.72	0.41
	TBK1	TANK binding kinase 1	−0.55	0.68
	MMP13	Matrix metallopeptidase 13	−0.46	0.71
	ANAPC5	Anaphase promoting complex subunit 5	−0.34	0.65
	TRADD	TNFRSF1A associated via death domain	−0.33	0.87
	SRSF1	Serine/arginine-rich splicing factor 1	−0.33	0.60
	ELAVL1	ELAV like RNA binding protein 1	−0.32	0.72
	CASP8	Caspase 8	−0.31	0.75
	TRAF5	TNF receptor-associated factor 5	−0.30	0.85
	IKBKB	Inhibitor of nuclear factor kappa-B kinase subunit beta	−0.28	0.72
	TAB3	TGF-beta activated kinase 1 (MAP3K7) binding protein 3	−0.14	0.89
	MAPK12	Mitogen-activated protein kinase 12	−0.07	0.94
	MAPK1	Mitogen-activated protein kinase 1	−0.07	0.94

FC = Fold change; FDR = false discovery rate.

## Data Availability

All results are included in the manuscript and its related Appendix A. All other inquiries can be directed to the corresponding author.

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
