# Peer review of "Identification of Critical Immune Regulators and Potential Interactions of IL-26 in Riemerella anatipestifer-Infected Ducks by Transcriptome Analysis and Profiling"

_microorganisms, 2024, doi:10.3390/microorganisms12050973_

Round 1

Reviewer 1 Report

Comments and Suggestions for Authors

Dear authors

I hope all of you are always fine. Regarding the revision of the manuscript No. microorganisms-2965742, entitled “Identification of Critical Immune Regulators and Potential Interactions of IL-26 in Riemerella anatipestifer–Infected Ducks by Transcriptome Analysis and Profiling”. It is really an interesting research discussing the host-pathogen interactions clarifying the riemerellosis pathogenesis; however, some comments should be replied.

Comments:

1-  The origin and full data about the Riemerella anatipestifer serotype 1 used for experimental infection should be indicated (including its accession number if present).

2-  Why did you use a very low dose of infection 500 or 5x102 CFU/200 ul?

3-  The referred research Fernandez et al., 2016 (16) used a very high dose of infection 5x107 that induced 80% mortality. Does the difference in dose could change the results of innate immune response including all up- or down-regulated cytokines and ILs?

4-  What was the aim of the other independent experiment applied? Where are its results?

5-  In Materials and Methods; 2.1. Animals and Ethics Statement: change food to feed.

Author Response

We sincerely appreciate the editor and reviewers for their valuable time, and insightful recommendations for enhancing our manuscript. The authors have diligently considered the comments, and below are our point-for-point responses. We trust these will align with your expectations and contribute to the further improvement of our manuscript.

Reviewer’s comments 1: 

Dear authors

I hope all of you are always fine. Regarding the revision of the manuscript No. microorganisms-2965742, entitled "Identification of Critical Immune Regulators and Potential Interactions of IL-26 in Riemerella anatipestifer–Infected Ducks by Transcriptome Analysis and Profiling". It is really an interesting research discussing the host-pathogen interactions clarifying the riemerellosis pathogenesis; however, some comments should be replied.

Comments:

1.) - The origin and full data about the Riemerella anatipestifer serotype 1 used for experimental infection should be indicated (including its accession number if present).

Response: We appreciate the reviewer's comment and have made the necessary revisions. The following sentence has been added and revised in lines 111-113: 'The bacteria used in this study was isolated from a commercial duck farm in Changwon, Gyeongnam Province, Korea, and was confirmed and serotyped at Chonbuk National University.'

2.)-  Why did you use a very low dose of infection 500 or 5x10^2 CFU/200 ul?

Response: We would like to clarify our choice of the infection dose. In a previous paper (Fernandez et al., 2016, Developmental and Comparative Immunology), ducks infected with 5 x 10^7 CFU of Riemerella anatipestifer (RA) serotype 1 had an 80% mortality rate. As RA serotype 1 is passaged in vitro and in vivo, RA serotype 1 becomes increasingly virulent. As a result, previously administered doses of 5x10^7 RA 1 resulted in a 100% mortality rate in infected ducks.

Hence, a series of pre-trials with different doses of RA serotype 1 isolate was used in the present study to determine the LD50 and a dose that will not cause 100% mortality. From these trials, we determined that an infection dose of 500 was appropriate. We have an ongoing study with the whole genome sequence of the isolate to determine potential differences in the virulence of the RA serotype 1 isolate used in this study. 

RA serotype 1 is considered one of the most pathogenic serotypes of RA and typically exhibits variable virulence. A recent study reported different LD50 for various RA serotype 1 isolates (3.5 x 10^3, 5x10^7, 1 x 10^8) (Liang et al., 2024).

Reference: Liang, Z. M., Li, H., Yang, D. H., Yin, L. J., Wu, Y. Y., Liu, J. F., & Zhou, Q. F. (2024). A novel bivalent inactivated vaccine for ducks against based on serotype distribution in southern China. Poultry Science, 103(3). https://doi.org/ARTN 10342710.1016/j.psj.2024.103427

3-  The referred research Fernandeinet al., 2016 (16) used a very high dose of infection 5x107 that induced 80% mortality. Does the difference in dose could change the results of innate immune response including all up- or down-regulated cytokines and ILs?

Response: We appreciate the reviewer's comment and would like to provide a more detailed explanation. While different doses may affect infection outcomes, the key cytokines IL-17A and related IL-17F, previously found to be implicated in RA infection, were consistently upregulated despite the difference in dose challenge used in this study. These results suggest that this experiment yielded quite similar infection outcomes. 

4-  What was the aim of the other independent experiment applied? Where are its results?

Response: The samples used for histopathological examination and RNA-seq analysis were collected simultaneously per group but were from different ducks. The following sentence has been revised in lines 125-126: “Additionally, spleens were collected from three ducks per group at 4 dpi for subsequent gene expression analysis by RNA-seq.”

 5-  In Materials and Methods; 2.1. Animals and Ethics Statement: change food to feed.

Response: The error indicated in the manuscript has been rectified in the revised version in Line 104.

Reviewer 2 Report

Comments and Suggestions for Authors

This manuscript studies the participation of immune –related genes in duck infections by   Riemerella anatipestifer (RA).  This study is aimed to understand the signaling and cellular network mechanisms underlying the host immune response during infection. Transcriptome analysis and profiling technics were applied to spleen samples from both infected and uninfected duckling.  The present study represents a continuation of previous studies by these authors, who found that upregulated IL-17A expression participated in the pathogenesis of this disease, but this mechanism was not linked to IL-23.   . The obtained results identify a possible interaction of IL-26 with IL-17; IL-26 is presented as a new upstream cytokine that could potentially regulate IL-17 in RA infection. Also, this study recognizes several pathways and genes related to the host immune response in RA-infected ducks. These findings would contribute to better understanding of RA infection and immunotherapy treatments; of great practical interest given the important economically impact of the disease on domestic fowl species worldwide.

 The introduction adequately describes the previous studies on which the study is based, many of which have been carried out by the authors of this work themselves. Thus, the novelty of the present study is clearly justified within the field of work of these researchers. Objectives were clear and well described. Methods have been described in detail and were adequate to objectives. Conclusions are very well based on results.

 In general, quality of tables and images are good, but I made some specific comments that must be addressed (see below). Supplementary materials are very interesting and contribute to better understanding of the manuscript.

  References list includes a total of 36 references (72.2%, 26/36, from the last 10 years; 41.7%, 15/36, from the last 5 years). 

 Specific comments:

-                     In Animal infections section it is said: Twenty 2-week-old ducks were injected ….. A similar procedure was followed for the uninfected negative control group (dNC). Please, indicate the number of ducklings in the control group.

-                     In Results section it is said: 3.4. GO Classification and Function Analysis of Upregulated DEGs Showed Significant Association with the Response to External Stimuli and Cytokine-Mediated Signaling Pathways. However, this subsection also includes information for Downregulated DEGs, associated with other pathways; please modify this title.

-         The letters in figure 3B are correct but in figures 3D and 3F they are inverted, as if reflected in a mirror.

-          Figure 5 is difficult to read; could its quality be improved?

-         About table 4 it is said: Collectively, 42.85% (45/105) of the genes associated with Th17 differentiation were expressed in RA-infected samples (Table 4).However, there is 46 genes in this table. Also, it is said about table 4: According to the RNA-seq results, 22 (23.91%) and 15 (16.30%) genes were upregulated and downregulated, respectively. But in this table 23 upregulated genes are presented. Please review these data.

-          In conclusions sections it is said: In accordance with previous reports, RNA-seq results equivocally demonstrated significantly upregulated expression of both IL-17A and IL-17F in RA infection. Do you really mean equivocally?.

-          Reference 31: this reference is incomplete, there is no journal number  neither pages

Author Response

We sincerely appreciate the editor and reviewers for their valuable time, and insightful recommendations for enhancing our manuscript. The authors have diligently considered the comments, and below are our point-for-point responses. We trust these will align with your expectations and contribute to the further improvement of our manuscript.

Reviewer’s comments 2: 

This manuscript studies the participation of immune –related genes in duck infections by   Riemerella anatipestifer (RA).  This study is aimed to understand the signaling and cellular network mechanisms underlying the host immune response during infection. Transcriptome analysis and profiling technics were applied to spleen samples from both infected and uninfected duckling.  The present study represents a continuation of previous studies by these authors, who found that upregulated IL-17A expression participated in the pathogenesis of this disease, but this mechanism was not linked to IL-23.   .The obtained results identify a possible interaction of IL-26 with IL-17; IL-26 is presented as a new upstream cytokine that could potentially regulate IL-17 in RA infection. Also, this study recognizes several pathways and genes related to the host immune response in RA-infected ducks. These findings would contribute to better understanding of RA infection and immunotherapy treatments; of great practical interest given the important economically impact of the disease on domestic fowl species worldwide.

 The introduction adequately describes the previous studies on which the study is based, many of which have been carried out by the authors of this work themselves. Thus, the novelty of the present study is clearly justified within the field of work of these researchers. Objectives were clear and well described. Methods have been described in detail and were adequate to objectives. Conclusions are very well based on results.

 In general, quality of tables and images are good, but I made some specific comments that must be addressed (see below). Supplementary materials are very interesting and contribute to better understanding of the manuscript.

  References list includes a total of 36 references (72.2%, 26/36, from the last 10 years; 41.7%, 15/36, from the last 5 years). 

 Specific comments:

1.) -In Animal infections section it is said: Twenty 2-week-old ducks were injected ….. A similar procedure was followed for the uninfected negative control group (dNC). Please indicate the number of ducklings in the control group.

Response: The indicated suggestion has been added to the revised manuscript at L:123 as "...uninfected negative control group (dNC, n=20)."

2.) -In Results section it is said: 3.4. GO Classification and Function Analysis of Upregulated DEGs Showed Significant Association with the Response to External Stimuli and Cytokine-Mediated Signaling Pathways. However, this subsection also includes information for Downregulated DEGs, associated with other pathways; please modify this title.

Response:  The section title has been modified in L:250-252 as "3.4. GO Classification and Function Analysis of DEGs Showed Significant Association with the Response to External Stimuli and Cytokine-Mediated Signaling Pathways for upregulated DEGs while extracellular matrix and collagen trimer for downregulated DEGs".

3.)- The letters in figure 3B are correct but in figures 3D and 3F they are inverted, as if reflected in a mirror.

Response: We appreciate the reviewer's comment and corrected the error.

4.)- Figure 5 is difficult to read; could its quality be improved?

Response: We have made the necessary improvements in the revised manuscript for Figure 5. 

5.)-About table 4 it is said: Collectively, 42.85% (45/105) of the genes associated with Th17 differentiation were expressed in RA-infected samples (Table 4). However, there is 46 genes in this table. Also, it is said about table 4: According to the RNA-seq results, 22 (23.91%) and 15 (16.30%) genes were upregulated and downregulated, respectively. But in this table 23 upregulated genes are presented. Please review these data.

Response: We appreciate the reviewer's comment, and the data has been reviewed, and the sentences in Lines 369-377 have been revised to avoid confusion as, " In the KEGG database, 105 genes were indicated as related to Th17 cell differentiation (ID: KEGG04659). In the KEGG gene list, 30 upregulated (28.57%) and 15 (14.28%) downregulated genes were present in the spleens of RA-infected ducks (Table 4). More specifically, seven upregulated genes were DEGs. Therefore, 42.85% (45/105) of the genes associated with Th17 differentiation was expressed in RA-infected samples (Table 4). Conversely, 92 genes were related to IL17 signaling (ID: KEGG:04657) in the KEGG database. According to the RNA-seq results, 22 (23.91%) and 15 (16.30%) genes were upregulated and downregulated, respectively (Table 5). Of the upregulated genes, four were DEGs. Accordingly, 40.21% (37/92) was expressed during RA infection (Table 5)." 

6.)-In conclusions sections it is said: In accordance with previous reports, RNA-seq results equivocally demonstrated significantly upregulated expression of both IL-17A and IL-17F in RA infection. Do you really mean equivocally?.

Response: We appreciate the reviewer's comment and have made the appropriate correction on the error in the revised manuscript in Lines 441-442 as, "In accordance with previous reports, RNA-seq results unequivocally demonstrated significantly upregulated expression of both IL-17A and IL-17F in RA infection."

7.)- Reference 31: this reference is incomplete; there is no journal number  nor pages

Response: The indicated error was corrected in the revised manuscript as "Shiri AM, Zhang T, Bedke T, Zazara DE, Zhao L, Lucke J, Sabihi M, Fazio A, Zhang S, Tauriello DVF et alIL-10 dampens antitumor immunity and promotes liver metastasis via PD-L1 inductionJ Hepatol 2024, 80:634-644." 
